# The Role of *Escherichia coli* Autotransporters in Urinary Tract Infections and Urosepsis

**DOI:** 10.3390/ijms26199760

**Published:** 2025-10-07

**Authors:** Beata Krawczyk, Paweł Wityk

**Affiliations:** 1Department of Biotechnology and Microbiology, Faculty of Chemistry, Gdańsk University of Technology, 80-233 Gdańsk, Poland; 2Department of Biopharmaceutics and Pharmacodynamics, Medical University of Gdańsk, 80-210 Gdańsk, Poland; pawel.wityk@gumed.edu.pl

**Keywords:** *E. coli*, Urinary Tract Infection, sepsis, autotransporters, virulence, therapy

## Abstract

Urinary tract infections (UTIs) caused by Uropathogenic *Escherichia coli* (UPEC) strains are among the most common bacterial infections in humans, causing cystitis, pyelonephritis and, in the absence of appropriate treatment, sepsis. Effective therapies and preventive strategies are still lacking, which highlights the need to better understand UPEC virulence mechanisms. Herein, we describe the role of three groups of bacterial autotransporters (ATs): serine protease autotransporter (SPATE), trimeric autotransporter adhesins (TAA), and autotransporter adhesin AIDA-I, and their possible contribution to the induction of UTI and urosepsis. AT, depending on the type, exhibits functions such as adhesion, serum resistance, hemagglutination, protease activity, biofilm formation and toxin activity. By summarizing the molecular functions of AT proteins, our review highlights their potential as targets for novel therapeutic and preventive approaches against UTIs and urosepsis.

## 1. Introduction

Sepsis is a life-threatening systemic infection associated with organ(s) dysfunction caused by a dysregulated host response to an infection. Depending on the geographical region, between 9% and 31% of severe sepsis cases are caused by urinary tract infection (UTI) [1,2]. Sepsis is most commonly caused by the Gram-negative bacteria *Escherichia coli* (*E. coli*), which is the main cause of hospital-acquired and non-hospital-acquired UTIs [3]. Due to the risk of sepsis and its long-term consequences, accurate diagnosis and early treatment of UTI is crucial [4]. In severe cases, bacteremia followed by sepsis can be fatal [2]. The most advanced form of sepsis is septic shock, which results in metabolic and circulatory problems leading to death [5].

Most *E. coli* strains isolated from the blood of patients that are hospitalized with sepsis originate from the upper urinary tract, and systemic infections caused by bacteria from the urinary tract are known as urosepsis [3]. Among the many virulence factors (fimbrial and afimbrial adhesins; iron uptake systems, e.g., siderophores, toxins) carried by uropathogenic *E. coli* is also a group of proteins called autotransporters (ATs) that play an important role in pathogenesis. The functions of *E. coli* ATs and their association with UTIs and sepsis are described in this article.

## 2. Autotransporters of *Escherichia coli* and Their Role in UTI

Autotransporter (AT) proteins are secreted via the Type V secretion system (T5SS), the most prevalent outer membrane transport pathway in Gram-negative bacteria [6]. This system enables the translocation of large protein domains to the bacterial surface or into the extracellular environment. AT proteins enable bacteria to colonize host tissues and develop infection [7].

All T5SS systems share a common feature: proteins are first exported across the inner membrane via the general protein secretory pathway (Sec), then translocated through a β-barrel domain embedded in the outer membrane. Their modular structure typically consists of three main domains: (i) N-terminal signal peptide—directs the protein through the Sec pathway across the inner membrane (ii) Passenger (α) domain—the functional effector portion, which may act as a toxin, adhesin, or protease; it is translocated across the outer membrane (iii) C-terminal β-barrel domain—anchors the protein in the outer membrane and forms a pore through which the passenger domain is secreted to the bacterial surface or extracellular environment [8]. This secretion mechanism is energy-independent and relies on the protein’s intrinsic ability to fold and translocate autonomously, hence the name “autotransporter”.

Autotransporters are key virulence factors exhibiting structural and functional diversity, which is reflected in several subtypes. The family of AT proteins secreted by *E. coli* strains is divided into three main groups, namely serine protease autotransporters of Enterobacteriaceae (SPATE), trimeric autotransporter adhesins (TAA) and AIDA-I-type autotransporter proteins [9]. AT proteins may have functions including adhesion, serum resistance, hemagglutination, protease activity, biofilm formation and toxin activity [10].

### 2.1. Serine Protease Autotransporters (SPATE)

The AT group of proteins includes SPATEs, the role of which is not fully understood [11]. The enzymatic activity of SPATE proteins is mediated by a conserved serine protease catalytic triad, consisting of aspartic acid (D), histidine (H), and serine (S). Within this triad, serine functions as the nucleophile, while aspartic acid stabilizes and activates histidine to facilitate peptide bond hydrolysis. Substitutions in any of these residues have been shown to completely abrogate proteolytic function across several SPATE family members [12,13].

SPATEs are classified according to their structural characteristics, falling into one of two categories: Class-1 or Class-2. Class 1 includes ATs with cytotoxic activity, whereas Class 2 proteins display immunomodulatory activity [14]. Their respective biological effects to UTI and urosepsis are outlined below.

#### 2.1.1. Class-1 SPATEs

*E. coli* Class-1 SPATEs cause tissue damage and cleave complement molecules and are among the virulence factors with a toxic function that is likely to be involved in the pathogenesis of sepsis. Class-1 proteins include the secreted autotransporter toxin (Sat), protein, which is involved in host–pathogen interactions and facilitate bacterial translocation into the bloodstream by damaging tissues [15].

Sat is one of the virulence factors of uropathogenic *E. coli* bacteria, exhibiting proteolytic activity [16] and induces a host immune response [17]. Sat cleaves a range of complement proteins (C2–C9), helping *E. coli* survive in bloodstream by evading complement-mediated killing. In a mouse model, deletion of *sat* cut lethality by ~50% in sepsis, underlining its role in pathogenesis [16]. Studies on cell cultures have shown that the Sat protease leads to cell elongation and detachment. Sat exerts cytopathic effects on the kidney, bladder and epithelial cells, causing vacuolization of bladder and kidney cells [17]. Studies in a mouse model also have shown that the Sat protein plays a crucial role in UTIs, contributing to kidney damage [18].

The next Class-1 AT protein is the extracellular serine protease (EspP), which has been implicated in cytotoxic activity, adhesion to host cells and biofilm formation. It is thought to be one of the major virulence factors of enterohemorrhagic *E. coli*. EspP can also cleave complement system molecules, which promotes the development and severity of hemolytic-uremic syndrome [15]. Studies in the Vero cell line confirmed the cytotoxic activity of the EspP protein, induction and damage to the cell cytoskeleton. Cells were observed to round and detach from the substrate, and intercellular junctions displayed damage [17]. Another protein in this group, EspC, is not fully understood but has been implicated as an enterotoxin [19]. Unlike EspP, it does not play a major role in adherence and invasion at the site of infection.

#### 2.1.2. Class-2 SPATEs

Class-2 proteins are mainly associated with mucosal colonization and immunomodulatory activities. Within the SPATEs family, the Pic protein (protein involved in intestinal colonization) plays a central role as a virulence factor in the enteroaggregative pathogenesis of *E. coli* [15]. The Pic protein secreted by *E. coli* mediates host immune system evasion by direct cleavage of complement system molecules. The Pic protease is responsible for the degradation of mucin, the chemical barrier of the epithelium, through the activation of mucinolytic enzymes. Studies in mice have shown that Pic production by *E. coli* promotes the development of sepsis, resulting in death within 12 h following infection. The cause of death from sepsis is thought to be the ability of the bacteria to survive in the bloodstream, the impaired expression of specific leukocyte surface molecules, and the effect on the induction of host inflammatory mediator production [20]. Vat protease (vacuolating autotransporter toxin) may also play an important role in urinary tract infections, allowing survival in the bloodstream [21]. In addition to its vacuolating and cytotoxic effects on epithelial cells, Vat has been implicated in promoting bacterial survival during systemic dissemination. According to Parham et al. (2005), Vat may facilitate survival in the bloodstream, potentially by modulating host immune responses or disrupting cellular barriers [21]. These properties suggest that Vat contributes not only to the colonization of the urinary tract but also to the progression of infection beyond the primary site. Other proteins belonging to the SPATE family include: PssA (Protease similar to Sat A), which may be responsible for cytotoxic effects on uroepithelial cells—similarly to Sat—by facilitating bacterial penetration into tissues and the bloodstream; Hbp enables *E. coli* to acquire iron from hemoglobin, a crucial function in iron-limited environments such as the urinary bladder, and thereby potentially supported bacterial survival [22]; and Pic-like proteases (e.g., Pic2, PicU), which exhibit immunomodulatory and mucinolytic activity—both essential for urinary tract colonization and evasion of host defenses [23]. Andersen et al. used quantitative proteomic profiling to characterize proteins expression of the UPEC strain UTI89 growing in human urine and when inside J82 bladder cells [24]. Whereas *pic* gene was expressed during urinary tract infection in a mouse model, and the protein was detected in the culture supernatants, indicating its secretion and potential role in virulence [25].

Several SPATEs from UPEC have been shown to disrupt epithelial integrity. Recently identified proteins TagB (Class-2) and TagC (Class-1) (encoded by tandem autotransporter genes located on a genomic island situated between the conserved *E. coli* genes *yjdI* and *yjdK*) and Sha (a serine protease with hemagglutinin activity, encoded on a virulence plasmid), have been implicated in the pathogenesis of urinary tract infections (UTIs) [26]. The *tagB* and *tagC* genes have also been identified in multiple multidrug-resistant clinical isolates, including members of the pandemic ST131 lineage (e.g., *E. coli* JJ1877) as well as other CTX-M-producing strains isolated from urinary tract infections and sepsis cases [26]. According to studies by Habouria et al. (2019), *tagB* and *tagC* were present in 10% of UTI isolates and 4.7% of avian *E. coli* [9]. The *sha* gene was present in 1% of UTI isolates and 20% of avian *E. coli.* These autotransporters exhibit cytopathic effects on bladder epithelial cells, which constitute a critical barrier against uropathogens. The functions of TagB, TagC, and Sha interact with bladder epithelial cells and define the role of their serine protease activity [26]. All three proteins were internalized by host cells and induced actin cytoskeletal rearrangement. In addition, Sha and TagC degraded mucin and gelatin, respectively. Mutation of the serine residue within the catalytic triad abolished internalization, cytotoxicity, and proteolytic activity, without affecting secretion. These results suggest that the catalytic activity of these SPATEs is essential for host cell entry and epithelial disruption.

In clinical *E. coli* isolates obtained from patients with pyelonephritis and urosepsis, the *sinH* gene was found to be highly prevalent [27,28,29]. The *sinH* gene is located within a horizontally acquired genomic island called the *ila* locus and is co-transcribed with the downstream gene *sinI* [27]. Recent work suggests that SinH shares a similar structural and evolutionary history with intimin and invasin as virulence-associated bacterial outer membrane proteins. The intrinsic ability to mediate its translocation across the inner membrane, periplasm, and outer membrane via the type V secretion system allows the protein to be displayed on the bacterial surface, where it can interact directly with host tissues. Functional studies have demonstrated that SinH can play a multifaceted role in host–pathogen interactions. It contributes to bacterial adhesion to epithelial surfaces, promotes autoaggregation, and facilitates invasion of host cells—each of which is a critical step in the progression of infection from colonization to systemic dissemination. SinH may contribute to bloodstream survival and is associated with increased risk of sepsis in murine models; in knockout strains, reduced survival in blood has been demonstrated, supporting a potential causal role. These virulence-related properties suggest that SinH may enhance the ability of pathogenic *E. coli* to evade host immune responses and establish infection in extraintestinal niches such as the urinary tract, bloodstream, and central nervous system [28].

### 2.2. Trimeric Autotransporter Adhesins (TAAs)

TAAs are a family of homotrimeric afimbrial adhesins belonging to important virulence factors for many Gram-negative bacterial pathogens. TAAs have many biological functions, such as the ability to mediate adhesion to eukaryotic cell surfaces or extracellular matrix proteins, through a so-called passenger domain that is exposed on the cell surface [30]. TAAs consist of short 70–100 amino acid C-terminal membrane-anchored domains that form a trimer, followed by a full-length pore (β-barrel) that facilitates translocation of the passenger domain to the cell surface [31,32,33]. Several types of TAAs from *E. coli* have been characterized. The UpaG protein, which affects bacterial aggregation, is involved in biofilm formation and adhesion to fibronectin, laminin and bladder epithelial cells [6]. These functions are critical for establishing infection in the urinary tract environment. Although deletion of the *upaG* gene did not significantly reduce bladder colonization in a murine UTI model using strain CFT073, the gene is frequently present in clinical isolates and has been associated with enhanced virulence traits. The autotransporter protein UpaG, as described by Krawczyk et al. (2025), was involved in 94% urosepsis isolates, and belonged to phylogenetic groups B2 and D [34]. Importantly, UpaG has also been shown to facilitate bacterial aggregation and biofilm formation, which are critical for persistence in the urinary tract environment and for resistance to host immune responses and antibiotic treatment [6].

Beyond these canonical autotransporters, another notable group comprises the Eib (*Escherichia coli* immunoglobulin-binding) proteins. Eib proteins form a family of surface molecules that interact with the Fc fragment of IgG—and in some cases IgA—in a nonimmune manner, thereby interfering with opsonization. They also confer serum resistance, enabling survival in the presence of complement [35,36]. These properties suggest an important role in systemic infection. In sepsis or during dissemination from a primary site such as urinary tract infection (UTI), Eib-mediated Fc binding and serum resistance may promote bacterial survival and persistence. Although direct clinical evidence linking *eib* carriage with UTI or sepsis isolates remains limited, the molecular mechanisms established in vitro provide strong biological plausibility. Systematic screening of clinical ExPEC isolates for *eib* and correlation with patient outcomes will be essential to clarify their contribution to human disease. In addition to UpaG and Eib, the TAA group in *E. coli* includes the Saa protein. All of these share a common C-terminal region, but they differ significantly in size [33]. UpaG protein consists of 1778 amino acid residues, whereas Saa and Eib have between 392 and 535 amino acids [33]. Furthermore, there are several paralogs of the genes encoding the Eib proteins, many of which are encoded within the same *E. coli* strain.

### 2.3. Autotransporter Proteins of the AIDA-I Type

The largest group of adhesins identified in *E. coli* are the AIDA-I type ATs. These proteins play an important role in bacterial-induced pathogenesis by promoting colonization and invasion of host cells, as well as facilitating infection through the formation of bacterial aggregates and biofilms [7]. The AIDA-I type includes the AIDA-I, antigen 43 (Ag43), TibA, and UpaH proteins, TibA [7,30]. The involvement of the AIDA-I protein in *E. coli*-induced pathogenesis is based on the promotion of adherence, aggregation, biofilm formation and invasion [7,37]. This has a significant impact on the colonization and persistence of bacteria on the surface of host cells, and, thus, on the development of bacterial UTIs [38].

Another protein, Ag43 plays a pivotal role in mediating cellular self-aggregation, which constitutes the initial step in bacterial biofilm formation [10]. The ability of bacteria to form biofilms is closely associated with enhanced survival and persistence within the urinary tract environment, where biofilms protect bacterial communities from host defenses and antimicrobial agents [39]. Importantly, Ag43 has been detected within bacterial biofilms colonizing the bladder epithelium of patients suffering from chronic urinary tract infections (UTIs), highlighting its clinical relevance [40]. Several genetically distinct variants of the Ag43 protein have been identified among diverse *E. coli* strains, exhibiting significant sequence variability that translates into differential aggregation capacities and biofilm phenotypes [7]. Moreover, membrane vesicles (MVs) secreted by *E. coli* have been found to carry surface-exposed Ag43 molecules [41]. Recent findings by Zavan et al. (2025) demonstrated that MVs containing Ag43 markedly enhance bacterial adherence to other *E. coli* cells, may facilitate a greater number of interactions between bacteria and significantly promote biofilm formation [41]. Therefore, MVs containing Ag43 may enhance biofilm formation by increasing the overall bacterial biofilm biomass. This discovery reveals a previously unrecognized function for autotransporter-loaded MVs in facilitating bacterial persistence and pathogenesis. Notably, Ag43 is capable of interacting with homologous autotransporters from other pathogenic *E. coli* strains, and potentially even across different bacterial species, suggesting that MVs carrying these proteins may serve as important mediators of interspecies communication and coordination during multi-species biofilm development.

The AIDA-I-type AT adhesin is TibA, a multifunctional protein associated with multiple virulence phenotypes [7]. Studies have shown that deletion of the *tib* locus from the enterotoxigenic *E. coli* (ETEC) strain H10407 eliminates TibA production, and reduces the bacteria’s ability to adhere and invade epithelial cells by approximately 75%. These results suggest that the TibA protein has adhesin and invasin activity [42]. Moreover, it has been shown that the TibA protein has self-association properties (TibA-TibA intercellular interaction) and can mediate the autoaggregation of *E. coli* cells. In addition, TibA expression promotes and significantly facilitates biofilm formation by *E. coli* on abiotic surfaces, such as catheters [38,42]. TibA, like Ag43, facilitates epithelial colonization and protects bacteria from both the host immune response and antimicrobials [7], and thus may also have an impact on the development of UTIs.

UpaH is the largest AT-type AIDA-I protein and is particularly abundant in UPEC strains. It has been implicated in bladder colonization and biofilm formation [43,44]. Unlike the AT proteins mentioned above, UpaH does not promote bacterial aggregation, adherence to bladder epithelial cells or binding to extracellular matrix proteins. Nevertheless, expression of the UpaH AT protein strongly influences biofilm formation [7]. UpaH proteins differ significantly in the amino acid sequence of the passenger domain, resulting in different variants of this protein with different biofilm-forming properties [43,44]. A study in mice found that the presence of the UpaH protein gave the wild-type strain CFT073 a slight advantage over a strain with a deletion of the *upaH* gene [43,44,45].

### 2.4. From Experimental Models to Clinical Relevance

Distinguishing between in vitro, animal, and epidemiologic evidence is critical for clarity and for making honest translational claims. Most functional insights into autotransporters—such as adhesion, biofilm formation, and serum resistance—originate from in vitro studies or murine UTI models. While these systems provide valuable mechanistic understanding, their predictive value for human infection outcomes remains limited. For example, deletion of individual autotransporter genes (e.g., *upaH*) can attenuate virulence in the reference strain CFT073, yet similar deletions in other UPEC backgrounds yield divergent results, highlighting the influence of genetic context [28,45]. Likewise, protective effects of recombinant autotransporter domains (e.g., SinH) have been demonstrated in mice, but their breadth and durability across diverse clinical lineages remain uncertain [28,45,46,47].

Epidemiologic surveys complement these findings by revealing lineage-specific distributions of autotransporters, yet such associations do not directly equate to functional relevance in vivo. To date, there is little conclusive evidence that natural mutations in autotransporters correlate with reduced severity of human UTI or sepsis. Therefore, moving toward translational application requires integrative approaches that combine mechanistic studies, standardized animal models, and clinical isolate analyses linked to patient outcomes. Only by aligning these complementary evidence streams can the field make reliable claims about the vaccine potential of autotransporters.

Functional division of autotransporters with evidence type is included in Appendix A.

## 3. The Distribution of Autotransporters Differs Across *E. coli* Lineages

Meta-analysis studies have revealed the presence of 20 sequence types (STs) (ST131, ST69, ST10, ST405, ST38, ST95, ST648, ST73, ST410, ST393, ST354, ST12, ST127, ST167, ST58, ST617, ST88, ST23, ST117, and ST1193) in extraintestinal ExPEC strains [48]. When discussing functional studies, it is important to note that most mechanistic insights into autotransporter biology have been derived from canonical UPEC reference strains such as CFT073 (ST73, phylogroup B2) and UTI89 (ST95, B2). These strains have been extensively characterized in vitro and in murine UTI models, providing much of the current understanding of autotransporter-mediated adhesion, biofilm formation, and immune interactions [49]. In contrast, data from other ExPEC sequence types including the globally dominant ST131 and additional animal derived isolates are comparatively limited and often rely on genomic surveys or association studies rather than detailed functional experiments [28].

This distinction should be considered when extrapolating functional roles across lineages, as lineage-specific differences in autotransporter repertoire and sequence variation may significantly influence phenotype. The distribution of autotransporters is also non-random across *E. coli* phylogenetic groups and clonal lineages. Several studies have demonstrated that many autotransporter genes (including adhesins such as Ag43, AIDA-I, and lineage-associated antigens such as SinH) are enriched in ExPEC, particularly within phylogroup B2 and common UPEC sequence types, compared with commensal strains [49]. Ag43 and AIDA-I show more variable distributions in non-B2 lineages or avian pathogenic *E. coli* (APEC) [50,51]. Reported carriage rates in large collections typically reach ~70–90% for SinH among B2 isolates, ~40–60% for Vat and Sat in UPEC, and considerably lower frequencies in other phylogroups.

Taken together, these data highlight that antigen distribution is strongly lineage-dependent, which has important implications for estimating potential vaccine coverage. Canonical UPEC strains (CFT073, UTI89) share several autotransporters, suggesting potential for broadly protective vaccines, while pandemic and emerging lineages (ST131, ST69, ST405, ST648) exhibit more variable repertoires, indicating that multi-antigen formulations may be required. Functional studies in murine UTI and avian models support the protective potential of these proteins and highlight their relevance for both human and zoonotic ExPEC infections. Overall, autotransporters represent promising components for the rational design of next-generation ExPEC vaccines. Table 1 summarizes lineage-specific variation in autotransporter distribution.

## 4. Autotransporters as Vaccine Candidates

### 4.1. Autotransporters as Potential Molecular Targets for Vaccines

Given their diverse pathogenic functions, AT represent attractive vaccine targets against UPEC, *E. coli* sepsis, and other Gram-negative pathogens [52,53].

Inhibition of adhesion mediated by autotransporters such as SinH, UpaG/H, and Ag43 can reduce bladder colonization, biofilm formation, and epithelial invasion [6,30,42,43,44]. Similarly, neutralization of Sat, which exerts cytotoxic and proteolytic activity, may attenuate epithelial damage [12,54]. Targeting Hbp, an autotransporter hemoglobin protease involved in iron acquisition, could also impair bacterial survival within the urinary tract [22,53].

Functional studies of autotransporter genes through deletion, mutation, or altered expression have demonstrated their contribution to UPEC pathogenesis in murine models and cell-based assays, occasionally supported by clinical isolates. However, UPEC strains typically encode multiple autotransporters alongside other adhesins and toxins. Consequently, deletion of a single autotransporter rarely abolishes virulence, and observed effects are often subtle or tissue-specific (e.g., bladder vs. kidney) or detectable only in competition assays. The genetic background also influences phenotypes: for example, UpaH deletion reduces virulence in strain CFT073 but not in other UPEC isolates [43]. Furthermore, some autotransporters are subject to phase variation or complex regulation, as illustrated by UpaE, which is controlled by the invertible promoter element *ipuS* [55]. Evidence of attenuation in human infections caused by autotransporter mutants remains scarce, and natural mutations in clinical isolates have not been conclusively linked to reduced severity of urinary tract infections (UTIs).

Despite these limitations, several autotransporters are being actively investigated as vaccine antigens, with SinH and selected SPATEs (e.g., Sat) emerging as promising candidates. Administration of SinH-3 (the third Ig-like domain of SinH) as an immunogen in mice significantly reduced bladder colonization in an acute UTI model, lowering colony-forming unit (CFU) counts by up to 44-fold [28,29]. Anti-SinH-3 immunization also increased urinary IgG and IgA titres and enhanced protection against *E. coli* ST131 in both colonization and bacteremia models. Building on these findings, a “Dual-Hit” vaccine strategy combining SinH-3 with pro-HlyA (a toxoid antigen) in a subcutaneous formulation conferred strong protection against ExPEC ST95 (UTI89), increasing survival to 66.7% and reducing bacterial loads in organs by 5–8 log units [29].

Sat has also been highlighted as a potential vaccine candidate. Freire et al. (2022) showed that Sat exhibits immunomodulatory activity by cleaving multiple complement proteins across all pathways (C2, C3, C3b, C4, C4b, C5–C9, but not C1q), underscoring its role in bloodstream infection and sepsis [54]. In a murine sepsis model, deletion of *sat* reduced lethality by 50%, whereas complementation with active Sat partially restored virulence. These results suggest that Sat represents a viable target for vaccine development and anti-virulence therapies, providing an alternative approach to address the growing problem of antibiotic resistance.

Although individual autotransporter immunization can confer measurable protection, the effect is typically partial, and broader protection may require multivalent formulations. Other autotransporters, including UpaG, UpaH, and Ag43, have been studied functionally but have not yet been tested as vaccine antigens in vivo. Importantly, human data are still lacking. Current knowledge of autotransporters as molecular vaccine targets is summarized in Table 2.

### 4.2. Antigenic Variability and Immune Correlates of Autotransporter Antigens

The development of autotransporter-based vaccines faces several challenges. Autotransporters are attractive targets due to their surface localization and involvement in adhesion and virulence; however, their passenger domains display extensive sequence heterogeneity across subgroups (e.g., multiple AIDA-I/Ag43 variants differing in passenger length and sequence) [59]. Such diversity may restrict cross-protection, and the weak immunogenicity of certain proteins could further limit vaccine efficacy. Recent studies on SinH have shown that recombinant domains confer protection in mouse models against selected sequence types (e.g., ST131 and related lineages), yet the potential for antigenic escape in populations with high sequence variability remains. Consequently, multivalent or multi-epitope strategies, or the identification of highly conserved fragments, appear necessary [28]. For combined antigens, survival often is higher (e.g., ~70–80%) especially against mixed strain challenge [28].

Mucosal immunity represents an additional determinant of success in UTI vaccine development. Experimental evidence indicates that local secretory IgA, as well as urinary tract IgG, correlate with effective clearance of colonization in animal models, whereas systemic IgG responses alone do not consistently translate into bladder protection in the absence of mucosal immunity [60].

Translation into clinical application is further complicated by issues of administration route, adjuvant selection, and delivery systems. Mucosal delivery elicits stronger local responses but raises concerns about safety and antigen stability. Similarly, many potent mucosal adjuvants are unsuitable for human use, underscoring the need for safe alternatives. Promising strategies include nanofiber-based platforms, sublingual formulations, and intranasal or intravaginal delivery, which combine enhanced mucosal stimulation with acceptable safety profiles [61].

In summary, although challenges remain, autotransporters continue to represent compelling candidates for inclusion in multicomponent vaccines against urinary tract infections, provided that optimized adjuvants, conserved antigen selection, and effective mucosal delivery strategies are developed.

## 5. Summary

Urinary tract infections, particularly those caused by uropathogenic *E. coli*, remain a major global health concern due to their prevalence, potential for recurrence, and risk of progression to life-threatening urosepsis. In the absence of fully effective therapies and methods to prevent the development of urosepsis, there is a need to better understand the virulence mechanisms of uropathogenic *E. coli* strains. The pathogenicity of UPEC strains is driven by a wide array of virulence factors, among which autotransporter (AT) proteins play a key role. These surface-associated proteins, secreted via the Type V secretion system, contribute to bacterial adhesion, invasion, immune evasion, and biofilm formation—processes critical for colonization and systemic dissemination. UPEC has several AT proteins described in this paper, which are involved in epithelial cell invasion and biofilm formation. Some ATs are responsible for kidney damage and facilitate the crossing of the renal blood-bed barrier, leading to urosepsis, and mediate serum resistance, which increases the chances of bacterial survival in the patient’s blood. This review highlights the involvement of three major classes of AT proteins—SPATEs, trimeric autotransporter adhesins (TAAs), and AIDA-I-type autotransporters—in UTI pathogenesis and the progression to sepsis. SPATE proteins such as Sat, Vat, and Pic exhibit proteolytic and immunomodulatory functions, often directly linked to tissue damage and immune system disruption. TAAs like UpaG and Eib proteins facilitate adhesion and serum resistance, while AIDA-I-type proteins such as Ag43, TibA, and UpaH promote aggregation, biofilm formation, and epithelial colonization. Some of these ATs, including SinH and Ag43, are also under investigation as vaccine candidates due to their strong surface expression and specificity for pathogenic strains. Understanding the molecular mechanisms mediated by AT proteins will facilitate the development of targeted interventions to prevent and treat UTIs and their complications, including urosepsis. SinH and Sat are of particular interest as vaccine candidates because they are surface-exposed and appear to be largely restricted to pathogenic E. coli strains, including uropathogenic and other extraintestinal pathogenic lineages, while being absent or rare in commensal isolates.

The main groups of *E. coli* ATs with evidence of involvement in UTI and urosepsis are shown in Figure 1.

## Figures and Tables

**Figure 1 ijms-26-09760-f001:**
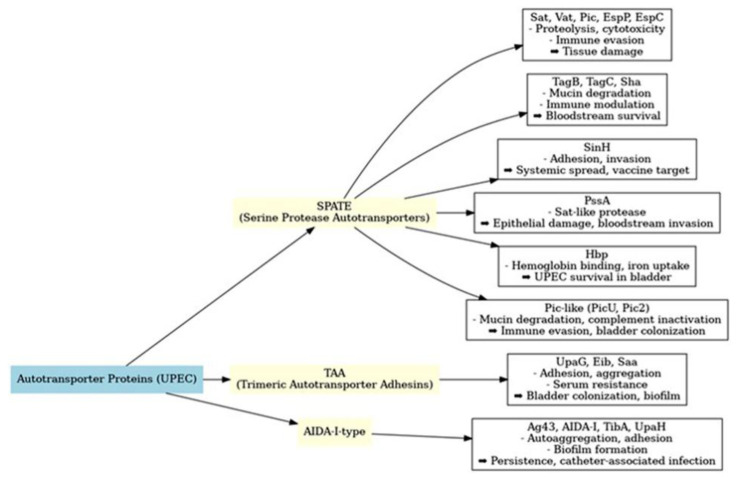
Functional classification of *E. coli* autotransporter (AT) proteins implicated in UTI and urosepsis.

**Table 1 ijms-26-09760-t001:** The distribution of autotransporters differs across *E. coli* lineages.

Strain/ST (Phylogroup)	Representative Status	Autotransporter(s) Studied	Type of Evidence	Conclusion	References
CFT073 (ST73, B2)	Canonical UPEC strain (human pyelonephritis)	Ag43, Pic, Vat, Tsh, Sat, Upa family	Extensive functional studies (adhesion, biofilm, immune modulation; murine UTI models)	CFT073 and UTI89 (canonical UPEC strains) share similar autotransporters (Ag43, Upa family, Sat), suggesting that vaccines targeting these proteins could protect against common urinary tract infections;	[49]
UTI89 (ST95, B2)	Canonical UPEC strain (human cystitis)	Ag43, Sat, Upa family	Functional studies (biofilm, intracellular lifestyle; murine UTI models)		[49]
ST131 (B2)	Pandemic ExPEC clone (human UTI, bacteremia)	SinH (mainly), Ag43 variants	Genomic surveys and vaccine-related studies; limited functional characterization	SinH and Ag43 variants, may be a key vaccine target for this widespread and clinically important clonal lineage	[28]
ST69, ST405, ST648 (various phylogroups)	Emerging ExPEC lineages	SPATEs (Vat, Sat), Ag43 alleles	Comparative genomics, prevalence studies; little functional data	Strains exhibit more diverse autotransporter repertoires. This antigenic heterogeneity suggests that a single-antigen vaccine may be insufficient, highlighting the need for multi-antigen formulations.	[51]

**Table 2 ijms-26-09760-t002:** Autotransporters as potential molecular targets for vaccines.

Protein/Autotransporter	T5SS Type	Mutant/Altered Expression/Effect	Models/Evidence Type	Role in UPEC	Evidence for Preventing UTI	Therapeutic/Vaccine Potential
SinH	Va	Recombinant SinH-based vaccine; knockout mutants	Murine UTI model (UTI89, CFT073), bacteremia protection studies; clinical/epidemiological data [28,56]	Hemagglutination, adhesion, cytotoxicity	Immunization reduces bladder colonization in mice	Promising vaccine candidate
Ag43	Va	Deletion mutants (Ag43a, Ag43b variants in CFT073)	In vitro assays; animal protection studies; clinical/epidemiological data [40]	Autoaggregation, biofilm formation, IBC formation	Blocking self-association reduces biofilm and aggregation	Potential target for inhibitors or blocking antibodies
UpaG	Vc	Mutagenesis; mutations in *upaG* or regulatory regions alter expression; *hns* deletion derepresses *upaG*	Biofilm assays; ECM binding (ELISA); epithelial cell binding (T24 bladder, Caco-2 intestinal cells) [57]	Adhesion, aggregation, biofilm formation	Mutants lacking UpaG show reduced adhesion and biofilm formation	Potential vaccine/therapeutic target by blocking adhesion
Sat	Va	*sat* gene deletion mutants; loss alters cytopathic and proteolytic effects	In vitro cytotoxicity (Vero, HK-2, CRL-1749, CRL-1573, HEK-293); mouse UTI model (kidney, bladder histopathology) [12]	Serine protease autotransporter; epithelial cytotoxicity; complement cleavage	Neutralization reduces epithelial damage in vitro	Vaccine or antibody-based therapeutic target
Hbp (Hemoglobin protease)	Va	Knockout mutants show attenuated virulence; catalytic-site mutants abolish protease activity	Animal models: BALB/c mice (abscess, UTI) and chickens [58]	Hemoglobin degradation; iron acquisition; survival in urinary tract	Loss of hemoglobin cleavage reduces virulence in vivo	Vaccine/therapeutic target to impair survival under iron limitation

## Data Availability

Not applicable.

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
