# Peer review of "The Role of Escherichia coli Autotransporters in Urinary Tract Infections and Urosepsis"

_ijms, 2025, doi:10.3390/ijms26199760_

Round 1
Reviewer 1 Report
Comments and Suggestions for Authors
Dear editor
Concerning the MS The role of Escherichia coli autotransporters in urinary tract infections and urosepsis
Comments
-The data requires more organizations, and more subtitles need to be added such as the Serine Protease Autotransporter need to be classified into subtitles.
-Also, among all the mentioned protein, there is no indications how the protein host pathogen interaction was detected.
-Is there any evidence on clinical cases or using mutant isolates to confirm the effects
- Does immunization against single transporter protein provide protection or against group p of proteins and what is the percentage of success.
Minor
All E. coli should be italic and G for gram should be capitalized.
Reviewer 2 Report
Comments and Suggestions for Authors
This is a timely, focused review of E. coli autotransporters (SPATEs, TAAs, AIDA-I family) in UTI and urosepsis. It gathers useful functional and vaccine-relevance data and will interest readers in pathogenesis and translational vaccinology. A few targeted edits will greatly sharpen the message.
Major points
- Many statements mix in-vitro, animal, and epidemiologic findings as if equal. Please label findings as (a) in vitro/cell data, (b) animal protection/knockout data, or (c) clinical/epidemiologic associations. e.g. for SinH and Sat say explicitly which results come from murine challenge studies vs human isolate prevlence, this makes translational claims honest and clearer.
- Autotransporter distribution varies by lineage (B2, D, ST131 etc). Briefly note when functions were shown in canonical UPEC strains (CFT073, UTI89) versus other ExPEC/animal isolats, and cite prevalence studies (or add a one-line percent if available). This affects vaccine coverage interpretation.
- Rephrase any “X causes bloodstream survival => sepsis” lines to be cautious (e.g., “may contribute” or “is associated with”). If there’s direct knockout/challenge evidence for bloodstream survival or mortality, highlight that explicitly, otherwise keep it associative.
- Add a short critical para on antigenic variability and immune correlates: mention sequence heterogenity of passenger domains (limits cross-protection), whether mucosal immunity (urine IgA/IgG) is likely needed vs systemic IgG, and practical delivery/adjuvant challenges for urinary vaccines.
Minor / editorial points
- Table 1: add columns for strain/model (in vitro / mouse) and evidence level (association vs functional vs vaccine protection).
- be consistent (gene italics vs protein caps — e.g., sinH vs SinH) and standarize SPATE names (Sat, Pic, Vat).
- Do a quick proofread to remove duplicated sentences and a couple small numeric mismatches.
Round 2
Reviewer 2 Report
Comments and Suggestions for Authors
Accepted in present form
Thanks for implementing my proposed comments.